# Unraveling the Bone–Brain Axis: A New Frontier in Parkinson’s Disease Research

**DOI:** 10.3390/ijms252312842

**Published:** 2024-11-29

**Authors:** Tingting Liu, Haojie Wu, Jingwen Li, Chaoyang Zhu, Jianshe Wei

**Affiliations:** Institute for Brain Sciences Research, Center for Translational Neurourology, Huaihe Hospital of Henan University, School of Life Sciences, Henan University, Kaifeng 475004, China; ltt0808@henu.edu.cn (T.L.); haojiewu@henu.edu.cn (H.W.); lijw0107@163.com (J.L.); zcy3015@126.com (C.Z.)

**Keywords:** Parkinson’s disease, bone–brain axis, inflammation, immunity, bone metabolism

## Abstract

Parkinson’s disease (PD), as a widespread neurodegenerative disorder, significantly impacts patients’ quality of life. Its primary symptoms include motor disturbances, tremor, muscle stiffness, and balance disorders. In recent years, with the advancement of research, the concept of the bone–brain axis has gradually become a focal point in the field of PD research. The bone–brain axis refers to the interactions and connections between the skeletal system and the central nervous system (CNS), playing a crucial role in the pathogenesis and pathological processes of PD. The purpose of this review is to comprehensively and deeply explore the bone–brain axis in PD, covering various aspects such as the complex relationship between bone metabolism and PD, the key roles of neurotransmitters and hormones in the bone–brain axis, the role of inflammation and immunity, microRNA (miRNA) functional regulation, and potential therapeutic strategies. Through a comprehensive analysis and in-depth discussion of numerous research findings, this review aims to provide a solid theoretical foundation for a deeper understanding of the pathogenesis of PD and to offer strong support for the development of new treatment methods.

## 1. Introduction

Parkinson’s disease (PD) is a neurodegenerative disorder primarily characterized by movement disorders, with clinical manifestations including tremor, muscle rigidity, bradykinesia, and more [1]. With the intensifying global aging trend, the incidence of PD has been rising steadily, posing a significant public health challenge. Currently, the etiology and pathogenesis of PD are not fully understood. Its defining characteristic is the irreversible loss of dopaminergic neurons in the substantia nigra and cholinergic neurons in the basal nucleus of Meynert within the central nervous system (CNS), coupled with progressive accumulation and aggregation of α-synuclein (α-syn) within the affected neurons, ultimately resulting in the formation of Lewy bodies (LBs) and Lewy neurites [2]. However, recent research has revealed that the onset and progression of PD may be linked to the interplay of multiple systems, with the concept of the bone–brain axis providing a novel perspective for understanding PD. Patients with PD exhibit heightened levels of α-syn in their bloodstream and the presence of LBs in the midbrain and enteric terminal nerves. On this basis, Figueroa and Rosen formulated a hypothesis that the aggregation of α-syn may disrupt the cellular dynamics of α-syn within bone cells, potentially contributing to bone dysfunction in PD patients [3]. Research on the bone–brain axis has unveiled complex signaling pathways between the skeletal system and the CNS. For instance, hormones and cytokines released by the bones can influence brain function, and vice versa. There is a certain correlation between skeletal diseases such as osteoporosis and osteoarthritis and cognitive decline, suggesting that bone health may have a profound impact on brain health [4,5]. In-depth investigation of the bone–brain axis relationship in PD is of great significance for revealing novel disease pathogenesis and identifying potential therapeutic targets.

The brain–bone axis encompasses a diverse array of signaling pathways, neuroendocrine factors, and molecular mediators that facilitate communication between the CNS and bone metabolism. The CNS exerts considerable control over bone remodeling through various neural pathways, including the sympathetic nervous system, hypothalamic neuropeptides, and neurotransmitters [6]. Conversely, bone-derived factors like osteocalcin (OCN) and fibroblast growth factor 23 (FGF23) have been demonstrated to influence brain function and behavior by crossing the blood–brain barrier (BBB) and modulating cognitive processes, memory, and emotional responses [7,8]. Bone morphogenetic proteins (BMPs) have been found to regulate the growth and differentiation of neuronal cells [9]. Abnormal expression of these molecules in the brain may be associated with the pathological processes of PD. Furthermore, bone marrow mesenchymal stem cells (MSCs) have demonstrated potential in repairing damaged neural tissue [10], further emphasizing the potential role of the brain–bone axis in neural regeneration and repair. In clinical practice, research on the brain–bone axis may provide new strategies for the prevention and treatment of PD. For example, improving bone health may help slow or halt the progression of certain PD [11]. Pharmaceutical treatments, lifestyle adjustments, and physiotherapy may all become new avenues for the future treatment of PD. Therefore, deeply exploring the mechanisms of the brain–bone axis not only aids in understanding the nature of these complex diseases but also provides hope for the development of new treatment methods.

As research progresses, it has been discovered that the interactions of the brain–bone axis may involve multiple signaling pathways and molecular mechanisms. For instance, stem cells in the bone marrow may affect brain function by secreting specific cytokines, while neural signals in the brain may regulate bone health by influencing bone metabolism. These findings suggest that PD may not solely be an issue within the brain but may also be related to systemic physiological changes. To better understand the role of the brain–bone axis in PD, researchers are conducting a series of experimental and clinical studies. They are attempting to simulate human disease processes through animal models to observe how changes in the brain–bone axis affect disease progression [12,13]. Meanwhile, clinical studies are collecting data on patients’ bone density, bone metabolism markers, and brain imaging to find correlations between abnormalities in the brain–bone axis and PD [14,15,16]. Additionally, based on research findings on the brain–bone axis, scientists are exploring new treatment strategies. For example, improving brain function by regulating bone metabolism [17] or using bone marrow stem cell therapy to repair damaged neural tissue [18]. The development of these treatment strategies not only provides new ideas for the treatment of PD but also brings hope for future clinical practice. In summary, research on the brain–bone axis offers a novel perspective for understanding PD. There is a close connection between bone and brain, which may play a crucial role in the occurrence and development of PD. This review will comprehensively and deeply explore the brain–bone axis in PD, aiming to provide new ideas and methods for its research and treatment.

## 2. Bone Metabolism and PD

### 2.1. Physiological Processes of Bone Metabolism

Bone metabolism refers to the continuous process of reconstruction and renewal of bones in the human body, which mainly involves two aspects: bone formation and bone resorption [19]. Bone formation is primarily handled by osteoblasts. Osteoblasts originate from MSCs and differentiate into osteoblasts under the stimulation of growth factors such as BMPs and transforming growth factor-β (TGF-β). Osteoblasts synthesize and secrete bone matrix components such as collagen and gradually mature the bone matrix through mineralization to form new bone tissue. In contrast, bone resorption is mainly mediated by osteoclasts. Osteoclasts derive from hematopoietic stem cells and differentiate into osteoclasts under the stimulation of factors such as macrophage colony-stimulating factor (M-CSF) and receptor activator of nuclear factor κB ligand (RANKL). Osteoclasts dissolve the bone matrix by secreting acidic substances and proteases, releasing minerals such as calcium and phosphorus, and allowing the bone tissue to be resorbed [20]. These two processes are in a dynamic equilibrium state in the human body, maintaining the normal structure and function of bones (Figure 1).

The physiological processes of bone metabolism are regulated by various factors, including hormones, cytokines, and mechanical stress. For example, parathyroid hormone (PTH) and vitamin D primarily promote bone resorption, while insulin-like growth factor-1 (IGF-1) and estrogen (ER) mainly promote bone formation [20,21]. Additionally, bone metabolism is influenced by factors such as genetics, age, gender, nutrition, and exercise. The balance of bone metabolism is crucial for bone health. If bone resorption exceeds bone formation, it can lead to osteoporosis, making bones fragile and prone to fracture; conversely, if bone formation exceeds bone resorption, it may result in osteopetrosis, making bones excessively hard and brittle [22]. Therefore, maintaining normal balance in bone metabolism is significant for the prevention and treatment of bone diseases such as osteoporosis.

### 2.2. Bone Metabolism Abnormalities in Patients with PD

Multiple studies have indicated the presence of bone metabolism abnormalities in patients with PD. Common manifestations include decreased bone mineral density and increased risk of fractures [23]. Some research has also found alterations in bone metabolism markers such as alkaline phosphatase (ALP), OCN, and osteoprotegerin (OPG) in PD patients [24,25,26]. These abnormalities suggest that bone metabolism may be affected during the pathogenesis of PD. The OPG/RANKL/RANK pathway not only plays a pivotal role in osteoclast activation, formation, and survival [27], but it also functions in the brain. In PD, the OPG/RANKL/RANK pathway may regulate the activation state of microglia [28]. For instance, RANKL can activate microglia and promote inflammatory responses, while OPG may inhibit microglial activation and reduce inflammation. This pathway may influence the release of inflammatory mediators such as tumor necrosis factor-α (TNF-α) and interleukin-1β (IL-1β) [29]. These inflammatory mediators exacerbate neuronal damage during the pathogenesis of PD. By regulating the OPG/RANKL/RANK pathway, the production of inflammatory mediators may be reduced, thereby exerting a protective effect on PD. The OPG/RANKL/RANK pathway may also have anti-apoptotic effects on neurons. OPG may protect dopaminergic neurons from damage by inhibiting apoptotic signaling pathways. Studies have shown that this pathway may be involved in neuroregeneration, regulation of oxidative stress, and clearance of α-syn [26].

PD patients often suffer from vitamin D deficiency [30]. Vitamin D plays a crucial role in calcium absorption and bone health. Vitamin D deficiency leads to inadequate calcium absorption, which affects the mineralization process of bones, making them fragile. Additionally, patients with PD have limited motor function and reduced activity levels, which also adversely affect bone metabolism. Exercise promotes bone formation and maintenance, and its absence leads to decreased bone mass and increased fracture risk. Furthermore, PD patients often use medications such as dopamine agonists and anticholinergics, which may impact bone metabolism [31,32]. Abnormalities in the nervous system may alter the tone and function of skeletal muscles, further affecting bone health.

## 3. The Relationship Between Abnormal Bone Metabolism and PD

Abnormal bone metabolism is closely linked to PD. Studies have shown that patients with PD often exhibit motor impairments, which are intimately associated with bone metabolism. Specifically, impaired motor function in Parkinson’s patients leads to decreased physical activity, thereby affecting the normal metabolic processes of the bones and predisposing them to issues such as osteoporosis [33]. Furthermore, neurotransmitter imbalance is a prominent feature of PD, and this imbalance also exerts an influence on bone metabolism [34]. For instance, dopamine, a crucial neurotransmitter, plays a pivotal role in regulating bone health. The decline in dopamine levels in Parkinson’s patients may lead to disordered bone metabolism, further increasing the risk of osteoporosis [35]. Hormonal changes also play a significant role in the relationship between PD and abnormal bone metabolism. Patients with PD may experience fluctuations in hormone levels, such as changes in sex hormones and thyroid hormones, which directly impact bone formation and resorption, thereby influencing bone metabolism [36,37,38]. Lastly, inflammation and immune system abnormalities are common issues in Parkinson’s patients. Studies have indicated that chronic inflammation and immune system dysregulation can affect bone metabolism through various mechanisms, leading to osteoporosis or other bone problems [39,40]. The inflammatory response in Parkinson’s patients may exacerbate the process of bone destruction, thereby increasing the risk of fractures. MicroRNAs (miRNAs) play a crucial role in the PD-related bone–brain axis as an important class of gene expression regulators. In summary, the relationship between PD and abnormal bone metabolism is multifaceted, involving motor impairments, neurotransmitter imbalances, hormonal changes, the interplay between inflammation and the immune system, and miRNA functional regulation (Figure 2). Understanding these complex connections is crucial for developing effective prevention and treatment strategies.

### 3.1. Movement Disorders and Bone Metabolism

The movement disorders in PD patients may lead to a significant reduction in their physical activity, which further impacts the normal progression of bone metabolism [41]. In the context of chronic lack of exercise, osteoblast activity may decrease, while osteoclast activity relatively increases. This imbalance results in a gradual reduction in bone mass [42]. Additionally, movement disorders can also adversely affect the tensile effects on muscles and bones, thereby influencing bone growth and repair processes. This impact is not limited to the reduction in bone mass but may also include an increased risk of bone metabolic abnormalities such as osteoporosis [43]. Therefore, there is a close association between movement disorders and bone metabolism abnormalities in PD patients, which needs to be managed through appropriate exercise and treatment.

### 3.2. Neurotransmitter Imbalance and Bone Metabolism

PD is primarily caused by the degeneration and death of dopaminergic neurons in the substantia nigra, leading to a decrease in dopamine levels. Dopamine not only plays a crucial role in regulating motor function but also has a certain impact on bone metabolism [44]. Studies have found that dopamine can regulate the activity of osteoblasts and osteoclasts, affecting bone formation and resorption. Additionally, neurotransmitters such as norepinephrine (NE) may also participate in the regulation of bone metabolism [45]. In patients with PD, besides a significant decrease in dopamine levels, abnormalities may also occur in the levels of other neurotransmitters, such as serotonin (also known as 5-hydroxytryptamine (5-HT)) and acetylcholine (ACh). Serotonin is an important neurotransmitter that plays a vital role in mood regulation and sleep cycles. However, the impact of serotonin on bone metabolism cannot be ignored. Research shows that serotonin can regulate bone density by affecting the proliferation and differentiation of osteoblasts, thereby influencing bone health to a certain extent [46]. ACh, a neurotransmitter that plays a key role in neuromuscular transmission, may indirectly affect bone health through its role in skeletal muscle. ACh induces cell proliferation and reduces ALP activity in osteoblasts via nicotinic acetylcholine receptors (nAChRs) and muscarinic acetylcholine receptors (mAChRs) [47]. Furthermore, through neuromuscular transmission, ACh regulates muscle contraction and relaxation, indirectly affecting bone loading and stability, which may in turn have a certain impact on bone metabolism [48].

### 3.3. Hormonal Changes and Bone Metabolism

The relationship between hormonal changes and bone metabolism is particularly evident in patients with PD. Significant alterations in hormone levels may occur in the bodies of Parkinson’s patients, which subsequently have important implications for bone metabolism. For instance, ER is a crucial hormone that plays a vital role in maintaining bone density and health. However, in Parkinson’s patients, particularly female patients, ER levels often decline significantly. This decrease in hormone levels may lead to abnormalities in bone metabolism, thereby increasing the risk of osteoporosis and fractures [36]. Besides ER, other hormones such as PTH and vitamin D also play important roles in the bone metabolism process of Parkinson’s patients [49]. PTH is primarily responsible for regulating blood levels of calcium and phosphorus, while vitamin D aids in calcium absorption and bone health. In Parkinson’s patients, the levels and functions of these hormones may be affected, further impacting the balance of bone metabolism.

### 3.4. Inflammation and Immunity

Patients with PD exhibit inflammation and oxidative stress responses, which not only damage the nervous system but may also impact bone metabolism. Inflammatory factors released by inflammatory cells can enhance osteoclast activity while inhibiting osteoblast activity, leading to increased bone resorption [50]. In PD, bone tissue may be in a state of chronic inflammation. When bone tissue is damaged or abnormal, the immune system is activated [51]. Among them, the nuclear factor kappa-B (NF-κB) signaling pathway is a crucial immune-inflammatory-related signaling pathway that, upon activation, promotes the release of various inflammatory cytokines such as TNF-α, IL-1β, and interleukin-6 (IL-6). These cytokines not only affect bone metabolism, leading to bone loss and other conditions [52], but they can also reach the brain through blood circulation. In the brain, microglia are the primary immune cells in the CNS. Circulating inflammatory factors can activate microglia, transforming them into a proinflammatory phenotype [53]. Activated microglia further release large amounts of neurotoxic substances and inflammatory mediators, such as reactive oxygen species (ROS) and nitric oxide (NO), triggering neuroinflammation [54]. Neuroinflammation damages neurons, particularly dopaminergic neurons in the substantia nigra pars compacta of the midbrain, which is one of the main pathological features of PD [55]. Meanwhile, factors such as OCN secreted by osteoblasts may also play a role in immune regulation [56]. OCN can regulate the function of immune cells and may act as a potential messenger molecule in the bone–brain axis, influencing the immune microenvironment of the brain and thereby indirectly affecting the progression of PD.

Research has found that OCN can influence the polarization of macrophages [57]. Macrophages are classified into two types of polarized cells: classically activated (M1) and alternatively activated (M2). Under the stimulation of bacterial toxins and other factors, macrophages polarize into the M1 type, expressing and releasing proinflammatory factors and ROS, which promote the occurrence of inflammatory responses. Conversely, under the induction of specific cytokines, macrophages polarize into the M2 type, expressing anti-inflammatory factors that help resolve inflammation and promote tissue growth and repair. OCN can inhibit the expression and release of macrophage inflammatory factors (such as IL-6, TNF-α, etc.) while promoting the expression and release of macrophage anti-inflammatory factors (such as TGF-β, IL-10, Arg1, etc.), thereby regulating inflammatory responses in the brain and maintaining the balance of the brain’s immune microenvironment [58]. Studies have shown that OCN may inhibit the overactivation of microglia, reducing the occurrence of neuroinflammation and protecting neurons [59]. OCN can cross the BBB to act on neuronal cells, regulating the synthesis and release of various neurotransmitters (such as dopamine, 5-HT, γ-aminobutyric acid (GABA), etc.) [60,61,62]. These neurotransmitters not only participate in the transmission of neural signals but are also closely related to immune regulation in the brain. For example, 5-HT can regulate the activity of immune cells and the secretion of inflammatory factors, while GABA has an inhibitory effect on immune responses. By regulating the levels of these neurotransmitters, OCN indirectly affects the brain’s immune microenvironment. In cases of brain injury or disease, neural repair and regeneration are important physiological processes. OCN may promote neural repair and regeneration by stimulating the proliferation and differentiation of neural stem cells [63]. At the same time, OCN can also regulate the function of glial cells, promoting their secretion of neurotrophic factors and extracellular matrix components to support neural repair [64]. These effects contribute to restoring normal brain function and improving the brain’s immune microenvironment.

### 3.5. miRNA and Bone Metabolism

In PD, dysregulation of miRNAs’ expression can affect neuronal survival and function. For instance, miR-34a/b/c expression is decreased in the brain tissue of PD patients [65]. These miRNAs often target genes related to apoptosis, such as members of the Bcl-2 family. When miR-34a/b/c expression is downregulated, their inhibitory effect on pro-apoptotic genes is weakened, making neurons more susceptible to apoptosis. Additionally, their reduction promotes the expression and aggregation of α-syn, further impacting brain function [65,66]. Furthermore, miR-133b is specifically expressed in midbrain dopaminergic neurons and its expression decreases in PD patients. miR-133b can regulate genes related to the development and function of dopaminergic neurons, and its absence can lead to degeneration of these neurons. Some miRNAs are involved in regulating neuroinflammatory processes. In PD, abnormal activation of microglia triggers neuroinflammation [67]. miR-124, a highly expressed miRNA in the brain, inhibits the activation of microglia. When miR-124 expression decreases, microglia are more easily activated, releasing large amounts of inflammatory mediators such as TNF-α, IL-1β, and IL-6, which exacerbate neuronal damage [68].

In bone tissue, miRNAs play crucial regulatory roles in the function of osteoblasts and osteoclasts. For example, miR-204 is expressed in osteoblasts and inhibits the expression of osteoblast-related genes, such as Runt-related transcription factor 2 (Runx2) and BMP2. When miR-204 expression is upregulated, osteoblast differentiation and function are suppressed, leading to reduced bone formation [69]. Regarding osteoclasts, miR-223 can regulate osteoclastogenesis and osteoclast activity. In PD patients, inflammatory infiltration alters miR-223 expression, thereby affecting osteoclast function and disrupting the balance between bone resorption and bone formation [70]. In the bone–brain axis of PD, some miRNAs can simultaneously impact both the nervous and skeletal systems [71], potentially indirectly affecting bone metabolism by influencing neuroendocrine pathways. In the brain, the hypothalamic–pituitary–adrenal (HPA) axis is closely related to bone metabolism. miRNAs can regulate the synthesis and secretion of HPA axis-related hormones, subsequently influencing the function of cells in bone tissue.

## 4. The Role of Neurotransmitters and Hormones in the Bone–Brain Axis

### 4.1. Dopamine

The role of dopamine in the bone–brain axis is primarily manifested in its regulation of bone growth and metabolism. As a crucial neurotransmitter, dopamine not only plays a key role in the CNS but also holds significant importance in the skeletal system. Dopamine is one of the neurotransmitters that receives the most attention in PD research. In the CNS, dopamine primarily participates in regulating motor function. In patients with PD, the degeneration and death of dopaminergic neurons in the substantia nigra lead to decreased dopamine levels, resulting in motor disorders [72]. Studies have shown that dopamine can affect the proliferation, differentiation, and function of bone cells by binding to receptors on the surface of bone cells [73]. For example, during bone formation, dopamine can enhance the activity of osteoblasts, thereby accelerating the synthesis and mineralization of bone matrix. Additionally, dopamine can inhibit the activity of osteoclasts, reducing bone resorption and maintaining bone mass balance [44]. On the other hand, dopamine can also indirectly regulate bone metabolism by influencing hormone secretion. For instance, dopamine can regulate the secretion of hormones such as PTH and vitamin D [74], thereby affecting bone metabolism.

### 4.2. 5-HT

5-HT is a neurotransmitter widely distributed in the human body, playing a crucial role in regulating mood, sleep, appetite, and pain perception. It not only modulates mood and sleep in the brain but also exerts significant effects within the skeletal system. By binding to specific receptors on the surface of bone cells, 5-HT participates in the regulation of bone metabolism [75]. Studies have found that 5-HT can promote the proliferation and differentiation of osteoblasts while inhibiting the formation and activity of osteoclasts, thereby facilitating bone formation and inhibiting bone resorption to a certain extent. Furthermore, 5-HT is closely associated with the occurrence and progression of osteoporosis, and its mechanism of action in bone metabolism is increasingly becoming a research focus [76]. Recent research indicates a correlation between 5-HT and PD. Specifically, 5-HT interacts with the dopamine system in the brain, and dysfunction of the dopamine system is one of the primary pathological features of PD [77]. 5-HT indirectly regulates motor control and emotional responses by influencing the release and reuptake of dopamine. Central 5-HT can regulate the secretion of pituitary hormones through neuroendocrine pathways, and these pituitary hormones in turn influence bone metabolism [78,79]. When 5-HT levels decrease, this neuroendocrine regulatory mechanism is disturbed, potentially indirectly affecting bone metabolism. In the intestine, PD patients exhibit dysfunction of the enteric nervous system, and the intestine is the primary site of peripheral 5-HT synthesis [80]. Abnormalities in the enteric nervous system may affect the synthesis and release of 5-HT, subsequently influencing bone metabolism through the gut–bone axis [81]. Additionally, peripheral 5-HT can act on cells in bone tissue through blood circulation. When peripheral 5-HT levels or the signaling pathways change, they can indirectly affect bone metabolism [82].

### 4.3. NE

NE is an important neurotransmitter and hormone, and its role in the bone–brain axis cannot be overlooked. NE regulates bone metabolism primarily by binding to β-adrenergic receptors on the surface of bone cells [83]. Studies have shown that NE can promote osteoblast activity and increase bone matrix synthesis, while inhibiting osteoclast activity and reducing bone resorption. Additionally, NE regulates the differentiation direction of bone marrow MSCs by influencing the bone marrow microenvironment, thereby playing a significant role in bone formation and repair processes [84]. Furthermore, NE indirectly affects bone metabolism by influencing inflammatory and oxidative stress responses [85]. NE is a crucial neurotransmitter that plays a key role in stress responses. When the human body is under pressure, the secretion of NE increases, triggering physiological responses such as an accelerated heart rate and elevated blood pressure. Research indicates that the NE system may play a role in non-motor symptoms of PD, such as mood disorders, cognitive impairments, and sleep disturbances [86]. Moreover, noradrenergic neurons in certain regions of the brain may also be affected by the pathological processes of PD, further exacerbating patients’ symptoms. Due to dysfunction of the peripheral NE system in PD patients, the regulation of bone metabolism becomes imbalanced, leading to disruptions in the processes of bone formation and bone resorption [87]. In most cases, there is a relative increase in bone resorption and a relative decrease in bone formation, ultimately resulting in decreased bone mineral density. This reduction in bone mineral density significantly increases the risk of fractures in PD patients, severely impacting their quality of life and prognosis [88]. Besides decreased bone mineral density, the healing process after fractures in PD patients may also be affected by abnormalities in the peripheral NE system. Disturbances in the NE system may impact multiple aspects of fracture healing, including local inflammatory responses, cell proliferation and differentiation, and angiogenesis at the fracture site. For example, NE may interfere with the normal functional coordination between osteoblasts and osteoclasts at the fracture site, impeding the formation and remodeling of bone callus, thereby causing delayed fracture healing [89,90,91].

### 4.4. ER

ER is a vital sex hormone that plays a significant role in regulating bone metabolism. By enhancing the activity of osteoblasts and inhibiting the activity of osteoclasts, ER helps maintain bone density [92]. After menopause, women experience a decline in ER levels, leading to accelerated bone loss and an increased risk of osteoporosis [93]. Patients with PD, particularly female patients, often have lower ER levels, which may be associated with the development and progression of PD [94]. Some studies suggest that ER may have neuroprotective effects on PD. ER can protect dopaminergic neurons through various pathways, such as regulating the activity of antioxidant enzymes and inhibiting inflammatory responses [95].

### 4.5. PTH

PTH is a crucial hormone that plays a significant role in regulating calcium–phosphorus metabolism and bone metabolism. It can stimulate the activity of osteoclasts, enhancing bone resorption, while simultaneously promoting the activity of osteoblasts to increase bone formation [96]. The secretion of PTH is modulated by blood calcium levels; when blood calcium levels decrease, the secretion of PTH increases, thereby promoting the release of calcium from bones to elevate blood calcium levels [97]. Changes in PTH levels may occur in patients with PD, which are potentially linked to abnormal bone metabolism. Some studies suggest that PTH may be involved in the pathogenesis of PD. PTH can affect dopaminergic neurons by influencing calcium signaling and modulating inflammatory responses, among other pathways [49].

## 5. The Role of Inflammation and Immunity in the Bone–Brain Axis

### 5.1. The Role of Inflammation in Bone Metabolism

In patients with PD, neurological abnormalities may impact the regulation of the bone–brain axis, subsequently influencing bone metabolism through the Ras homolog family member A (RhoA)/Rho-associated coiled-coil-containing protein kinase (ROCK) signaling pathway. The RhoA/ROCK signaling pathway also plays a role in regulating the functions of osteoblasts and osteoclasts. Abnormal activation of this pathway may hinder osteoblast differentiation and activity, leading to decreased bone formation, while potentially enhancing osteoclast activity and increasing bone resorption [98]. The Notch signaling pathway is crucial for bone development and homeostasis. Bone cells primarily express Notch1 and Notch2, with lower levels of Notch3 [99]. Activation of Notch in osteoblasts, osteocytes, and stromal cells results in a decreased RANKL/OPG ratio and inhibition of M-CSF gene expression, consequently reducing their capacity to support osteoclastogenesis and bone resorption. Consequently, a deficiency in Notch1 indirectly promotes osteoclastogenesis by enhancing the ability of osteoblast lineage cells to elevate the RANKL/OPG expression ratio [99,100].

TNF-α, an important inflammatory cytokine, plays a complex role in bone metabolism. It inhibits osteoblast differentiation and function, reducing bone formation. TNF-α directly enhances RANKL expression in bone cells and promotes osteoclast formation, thereby increasing bone resorption [101,102]. IL-1, which includes IL-1α and IL-1β subtypes, also plays a significant role in bone metabolism. IL-1 stimulates osteoclastogenesis and activity, increasing bone resorption. Additionally, IL-1 inhibits osteoblast differentiation and function, reducing bone formation. Furthermore, IL-1 promotes inflammatory responses, exacerbating bone metabolism disorders [103]. IL-6, a multifunctional cytokine, is also crucial in bone metabolism. It stimulates osteoclastogenesis and activity, increasing bone resorption. Simultaneously, IL-6 inhibits osteoblast differentiation and function, reducing bone formation. Moreover, IL-6 participates in inflammatory responses and immune regulation, affecting the balance of bone metabolism [104].

Beyond these inflammatory cytokines, numerous other inflammatory factors are involved in regulating bone metabolism, such as interferon-γ (IFN-γ) and interleukin-17 (IL-17). These inflammatory factors can impact bone metabolism through various mechanisms, further exacerbating the occurrence and development of bone metabolic diseases [105]. Inflammation can also influence hormone secretion, indirectly regulating bone metabolism [106,107]. For instance, inflammation can inhibit the activation of vitamin D, lowering blood calcium levels, which stimulates PTH secretion and promotes bone resorption [96]. Additionally, inflammation affects the secretion of sex hormones such as ER and androgen (AR), thereby influencing bone metabolism [92].

### 5.2. The Role of Immunity in Bone Metabolism

T cells play a significant role in bone metabolism. IFN-γ secreted by Th1 cells can inhibit the differentiation and function of osteoblasts, promote the generation and activity of osteoclasts, and increase bone resorption [108]. IL-17 secreted by Th17 cells also promotes osteoclastogenesis and activity, leading to increased bone resorption [109]. In contrast, regulatory T cells (Treg) inhibit inflammatory responses and promote bone formation by secreting inhibitory cytokines such as IL-10 and TGF-β [110]. The role of B cells in bone metabolism is not fully understood. Some studies suggest that B cells participate in the regulation of bone metabolism by secreting antibodies and cytokines. For example, B cells can secrete RANKL, which promotes osteoclastogenesis and activity, increasing bone resorption [111]. Additionally, B cells interact with T cells to affect the balance of bone metabolism. Macrophages also play a crucial role in bone metabolism. In an inflammatory state, macrophages can be activated to secrete various inflammatory cytokines such as TNF-α, IL-1, and IL-6, which promote osteoclastogenesis and activity, increasing bone resorption [112,113]. Meanwhile, macrophages participate in the regulation of bone metabolism by phagocytosing cellular debris and pathogens in bone tissue. Abnormalities in inflammation and immunity in PD bone metabolism may impact the progression of PD. On the one hand, bone metabolism abnormalities may further impair patients’ motor function, exacerbating PD symptoms. On the other hand, inflammation and immune responses may further aggravate the neurodegeneration in PD by affecting the neuro-immune–bone axis.

## 6. Potential Therapeutic Strategies

The application of the bone–brain axis in the treatment of PD currently encompasses various methods, including exercise therapy, pharmacotherapy, nutritional intervention, stem cell therapy, and miRNA therapy strategy (Figure 3). Firstly, exercise therapy aids patients in improving motor function, enhancing muscle strength, and coordination through specific physical exercises and rehabilitation training. Secondly, pharmacotherapy is one of the primary means of treating PD, employing dopamine agonists, anticholinergic drugs, and other medications to alleviate symptoms and enhance patients’ quality of life. Additionally, nutritional intervention plays a significant role, with a balanced diet and nutritional supplements providing necessary nutritional support for patients and boosting their physical resilience. As an emerging treatment method, stem cell therapy holds promise for future PD treatment by transplanting stem cells to repair damaged nerve cells. Lastly, the success of miRNA biomarker research has the potential to lead to more precise disease diagnosis through less invasive methods. This holds particular significance in PD, where diagnoses are frequently made based on symptomatic examinations and cognitive assessments, rather than through more objective tests like blood tests. The combined application of these methods offers a comprehensive treatment plan for PD patients, helping to improve their symptoms and quality of life.

### 6.1. Exercise Therapy

Exercise therapy is one of the most important treatment methods for PD. Exercise can improve motor function and enhance the quality of life of PD patients. It also promotes bone metabolism, increases bone density, and reduces the risk of fractures [114]. Furthermore, exercise can regulate the secretion of neurotrophic factors, neurotransmitters, and hormones, thereby alleviating inflammation and oxidative stress responses [115]. For PD patients, it is crucial to choose appropriate types and intensities of exercise. Generally, aerobic exercises such as walking, jogging, and swimming can improve cardiorespiratory function and promote blood circulation, benefiting PD patients. Strength training exercises like weightlifting and push-ups can increase muscle strength and improve body stability, also providing certain benefits to PD patients. The intensity of exercise should be determined according to the patient’s physical condition and exercise capacity to avoid fatigue and injury due to overexertion.

### 6.2. Pharmacological Treatments

#### 6.2.1. Dopamine Replacement Therapy

Dopamine replacement therapy is one of the primary treatments for PD. It replenishes the deficiency of dopamine and alleviates motor symptoms in PD patients [116]. Additionally, dopamine replacement therapy may exert certain effects on bone metabolism. Some studies suggest that it can increase bone density and reduce the risk of fractures [117].

#### 6.2.2. Anti-Osteoporosis Drugs

Anti-osteoporosis drugs can enhance bone density and lower the risk of fractures [118]. For PD patients with concurrent osteoporosis, anti-osteoporosis drugs may be considered for treatment. Commonly used anti-osteoporosis drugs include bisphosphonates, hormone therapy, selective estrogen-receptor modulators, calcitonin, denosumab, and calcium and vitamin D supplementation.

#### 6.2.3. Anti-Inflammatory Drugs

Anti-inflammatory drugs can reduce inflammatory responses and may have therapeutic effects on PD patients. Furthermore, they may also exert certain effects on bone metabolism. Some studies have shown that anti-inflammatory drugs can decrease the risk of fractures and increase bone density. Since PD is an age-related disease, most PD patients are associated with musculoskeletal diseases that require long-term use of analgesics and anti-inflammatory drugs, such as non-steroidal anti-inflammatory drugs (NSAIDs). Therefore, NSAIDs can affect PD neuropathology in various ways. Inhibiting neuroinflammation and modulating immune responses with NSAIDs may be effective in preventing the progression of PD. The impact of NSAIDs on PD neuropathology can be either beneficial or harmful. Inhibition of cyclooxygenase-2 (COX2) by NSAIDs may halt the progression of PD. NSAIDs exert neuroprotective effects on the development and progression of PD neuropathology by modulating neuroinflammation [119].

#### 6.2.4. Antioxidants

Antioxidants can mitigate oxidative stress responses and may have therapeutic effects on PD patients [120]. Additionally, antioxidants may also have certain effects on bone metabolism. A diet rich in whole plant foods with a high antioxidant content, along with lifestyle changes that preserve antioxidants, may improve bone mineral density and reduce the risk of fragility-related fractures [121].

### 6.3. Nutritional Intervention

#### 6.3.1. Calcium and Vitamin D

Calcium and vitamin D are crucial nutrients for maintaining bone health. PD patients, especially those with concurrent osteoporosis, should appropriately supplement calcium and vitamin D to maintain bone density. The intake of calcium should be determined based on factors such as the patient’s age, gender, and physical condition. Generally, the daily calcium intake for adults should be between 800 and 1200 mg [122]. Similarly, the intake of vitamin D should also be tailored to individual factors, with a general recommendation of 400–800 IU per day for adults [123].

#### 6.3.2. Protein and Amino Acids

Protein and amino acids are important nutrients for maintaining bone health. PD patients should appropriately increase their intake of protein and amino acids to promote bone formation and repair. The protein intake should be determined based on factors such as the patient’s age, gender, and physical condition. Typically, the daily protein intake for adults should be between 0.8 and 1.2 g/kg of body weight [124].

#### 6.3.3. Other Nutrients

Apart from calcium, vitamin D, protein, and amino acids, PD patients should also appropriately supplement other nutrients, such as trace elements like magnesium, zinc, and copper, as well as antioxidants like vitamin C and vitamin E. These nutrients play vital roles in maintaining bone health and nervous system function.

### 6.4. Stem Cell Therapy

Stem cells are cells with the ability to self-renew and differentiate. They can be classified into two types: embryonic stem cells (ESCs) and adult stem cells. ESCs possess totipotency, meaning they can differentiate into any type of cell. In contrast, adult stem cells exhibit multipotency, capable of differentiating into specific types of cells. In the treatment of PD, commonly used stem cells include MSCs and neural stem cells (NSCs), among others [125].

The mechanisms of stem cell therapy in PD primarily encompass the following aspects: firstly, stem cells can differentiate into dopaminergic neurons, replacing damaged neurons and restoring the function of the nervous system. This helps improve motor symptoms in PD patients, such as tremor, stiffness, and bradykinesia [126]. Secondly, stem cells can secrete various growth factors and cytokines, such as brain-derived neurotrophic factor (BDNF) and nerve growth factor (NGF), promoting the survival and regeneration of surviving neurons, thereby further enhancing neural function [127]. Thirdly, stem cells can regulate the immune system and alleviate inflammatory responses [128]. Fourthly, stem cells can promote bone formation and repair, improving bone metabolism. They can differentiate into osteoblasts, facilitating bone regeneration and repair. This contributes to improving bone health in PD patients and reducing the risk of fractures [129]. Additionally, stem cells can secrete multiple growth factors, such as BMP and TGF-β, promoting the differentiation and activity of osteoblasts and enhancing bone formation [130]. By regulating the function of osteoclasts, stem cells can also influence the balance of bone metabolism [131]. For instance, MSCs can secrete factors that inhibit osteoclastogenesis, reducing bone resorption, while simultaneously promoting osteoblast activity and increasing bone formation. Regulating bone metabolism helps maintain the structure and strength of bones, improving bone health in PD patients.

Stem cell therapy for PD holds promising prospects but also faces several challenges. On the one hand, the safety and effectiveness of stem cell therapy need further validation. Risks such as immune rejection and tumorigenesis associated with stem cell therapy require rigorous safety assessments. On the other hand, the technical difficulty of stem cell therapy is high, necessitating improvements in stem cell differentiation efficiency and targeted transplantation capabilities. Furthermore, the high cost of stem cell therapy also limits its widespread clinical application.

### 6.5. MiRNA Therapy Strategy

#### 6.5.1. Targeted Regulation of miRNAs

For miRNAs that are dysregulated and play critical roles in the PD bone–brain axis, drugs can be designed to regulate their expression. For instance, for miRNAs that are downregulated in PD and have neuroprotective effects on neurons, nucleic acid-based drugs can be developed to increase their levels in the body. Antisense oligonucleotides (ASOs) or microRNA mimics can be designed to enhance the expression of specific miRNAs [132]. Conversely, for miRNAs that are overexpressed in PD and promote disease progression, miRNA inhibitors such as antagomirs can be used to reduce their expression levels, thereby mitigating their adverse effects on neurons and bone cells [133].

#### 6.5.2. Combination Therapy Strategy

miRNA-targeted therapy can be combined with existing PD treatment methods. For example, while using dopamine replacement therapy to treat motor symptoms in PD patients, we can regulate the miRNAs related to the bone–brain axis to improve patients’ bone health and delay neurodegeneration. This combined treatment strategy holds promise for enhancing the overall quality of life for PD patients.

#### 6.5.3. Development of Biomarkers

Due to the specific expression changes of miRNAs in the PD bone–brain axis, they can be utilized as biomarkers for early diagnosis and prognosis assessment of PD. By detecting the levels of specific miRNAs in blood, cerebrospinal fluid, or tissue samples, it can assist in the diagnosis of PD, especially in the early stages of the disease where traditional diagnostic methods may not be sufficiently sensitive. Furthermore, monitoring the dynamic changes in relevant miRNAs during the treatment of PD patients can evaluate treatment efficacy and predict disease progression, providing a basis for clinicians to adjust treatment plans [134,135].

## 7. Conclusions

The bone–brain axis in PD represents a complex network, where the bone and brain interact through neural, endocrine, and immune pathways. Factors such as abnormal bone metabolism, neurotransmitter and hormone imbalances, inflammation, and oxidative stress may play crucial roles in the pathogenesis and progression of PD. Therapeutic strategies targeting the bone–brain axis may offer new ideas and approaches for the treatment of PD. Future research needs to further delve into the mechanisms of the bone–brain axis, develop more effective treatments, and bring more hope to PD patients. Despite the current understanding of the bone–brain axis in neurodegenerative diseases and the theoretical potential of bone–brain axis-based therapeutic strategies, translating these strategies into clinical practice faces numerous challenges. Firstly, the interaction mechanisms of the bone–brain axis are extremely complex, involving multiple layers of regulation and signaling pathways, and our current understanding of it is not yet comprehensive and in-depth. Further basic research is needed to clarify the specific molecular mechanisms and key regulatory nodes of the bone–brain axis in the pathogenesis and progression of neurodegenerative diseases, providing a theoretical basis for precision medicine. Secondly, clinical research needs to address issues such as how to accurately assess the functional status and changes in the bone–brain axis, and how to determine the optimal timing, dosage, and duration of treatment. Currently, there is a lack of effective biomarkers and monitoring tools to reflect the dynamic changes in the bone–brain axis in real time, posing difficulties in formulating and adjusting clinical treatment plans. In addition, most bone–brain axis-based therapeutic strategies are still in the experimental or preliminary clinical application stages, and their long-term efficacy and safety require further verification and evaluation. Especially in combination therapy, the interactions and potential adverse reactions between different drugs or treatment methods need close monitoring.

Future research directions can include the following aspects: firstly, deeply exploring the connection between the bone–brain axis and the genetic and epigenetic mechanisms of neurodegenerative diseases to identify potential therapeutic targets and biomarkers. Secondly, utilizing advanced technological means, such as multi-omics analysis, bioinformatics, and imaging technologies, to comprehensively dissect the molecular networks and cellular communication patterns of the bone–brain axis in healthy and diseased states. Thirdly, conducting large-scale, multicenter clinical trials to validate the effectiveness and safety of bone–brain axis-based therapeutic strategies, optimize treatment protocols, and provide reliable evidence for clinical application. Fourthly, strengthening interdisciplinary collaboration to promote exchanges and cooperation among neuroscience, orthopedics, endocrinology, immunology, and other disciplines, jointly advancing the development and clinical translation of bone–brain axis research. In summary, the bone–brain axis in PD is a field worthy of in-depth study. Through research on the bone–brain axis, we can better understand the pathogenesis of PD and provide a theoretical basis for developing new treatment methods. Meanwhile, we should also pay attention to the bone health of PD patients, adopt comprehensive treatment measures, and improve their quality of life.

## Figures and Tables

**Figure 1 ijms-25-12842-f001:**
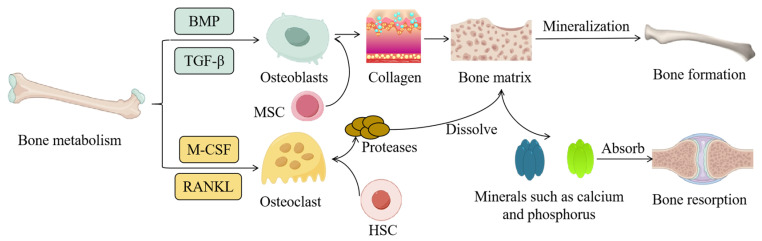
The physiological process of bone metabolism involves two main processes: bone formation and bone resorption. Bone formation is primarily mediated by osteoblasts. These osteoblasts differentiate under the stimulation of growth factors such as bone morphogenetic proteins (BMPs) and transforming growth factor-β (TGF-β). In contrast, bone resorption is mainly handled by osteoclasts. Osteoclasts differentiate in response to factors including macrophage colony-stimulating factor (M-CSF) and receptor activator of nuclear factor κB ligand (RANKL) [19,20].

**Figure 2 ijms-25-12842-f002:**
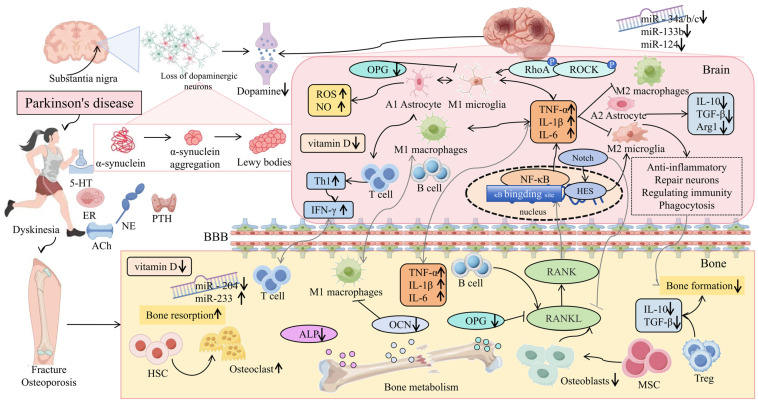
The relationship between PD and abnormal bone metabolism is multifaceted, involving movement disorders, neurotransmitter imbalances, hormone changes, and the interaction between inflammation and the immune system [33,34,35,36,37,38,39,40].

**Figure 3 ijms-25-12842-f003:**
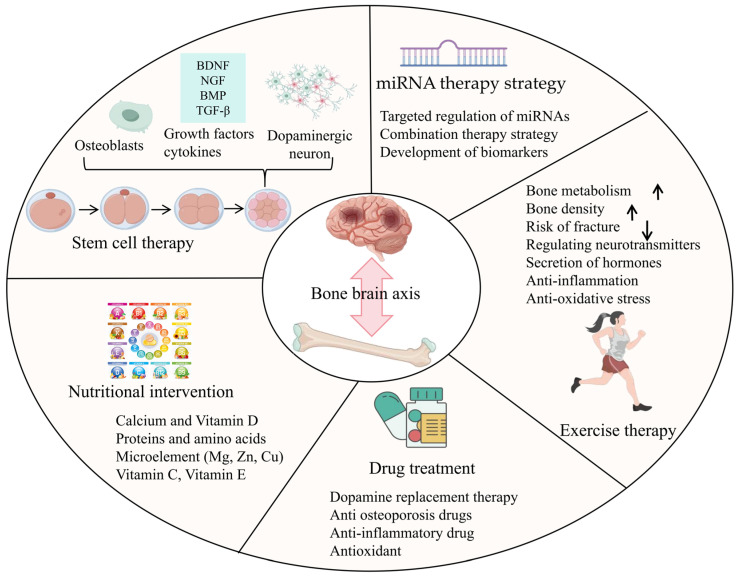
The application of the bone–brain axis in the treatment of PD mainly includes various methods, such as exercise therapy, drug therapy, nutritional intervention, stem cell therapy, and miRNA therapy strategy [114,115,116,117,118,119,120,121,122,123,124,125,126,127,128,129,130,131,132,133,134,135].

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
