# Peer review of "Unraveling the Bone–Brain Axis: A New Frontier in Parkinson’s Disease Research"

_ijms, 2024, doi:10.3390/ijms252312842_

Round 1

Reviewer 1 Report

Comments and Suggestions for Authors

The authors provide a comprehensive review on bone-brain axis in PD. The review is informative and interesting. 

However, just one concern on this review:

The miRNAs play a critical role in the bone-brain axis in PD, influencing both neurodegenerative processes and bone health. The authors should add a new section to discuss the roles of miRNA in bone-brain axis, including relevant pathogenesis and therapeutic strategies. 

Author Response

Thank you for your valuable feedback and suggestion. We appreciate your recognition of our comprehensive review on the bone-brain axis in Parkinson's disease (PD).

In response to your concern, we acknowledge that microRNAs (miRNAs) indeed play a pivotal role in the bone-brain axis, with significant impacts on both neurodegenerative processes and bone health in PD. To address this, we will incorporate a new section in our review to discuss the roles of miRNAs in the bone-brain axis. This section will cover relevant pathogenesis, mechanisms of action, and potential therapeutic strategies involving miRNAs.

In PD, dysregulation of miRNAs' expression can affect neuronal survival and function. For instance, miR-34a/b/c expression is decreased in the brain tissue of PD patients [65]. These miRNAs often target genes related to apoptosis, such as members of the Bcl-2 family. When miR-34a/b/c expression is downregulated, their inhibitory effect on pro-apoptotic genes is weakened, making neurons more susceptible to apoptosis. Additionally, their reduction promotes the expression and aggregation of α-synuclein, further impacting brain function [65, 66]. Furthermore, miR-133b is specifically expressed in midbrain dopaminergic neurons and its expression decreases in PD patients. miR-133b can regulate genes related to the development and function of dopaminergic neurons, and its absence can lead to degeneration of these neurons. Some miRNAs are involved in regulating neuroinflammatory processes. In PD, abnormal activation of microglia triggers neuroinflammation [67]. miR-124, a highly expressed miRNA in the brain, inhibits the activation of microglia. When miR-124 expression decreases, microglia are more easily activated, releasing large amounts of inflammatory mediators such as TNF-α, IL-1β, and IL-6, which exacerbate neuronal damage [68].

In bone tissue, miRNAs play crucial regulatory roles in the function of osteoblasts and osteoclasts. For example, miR-204 is expressed in osteoblasts and inhibits the expression of osteoblast-related genes, such as Runt-related transcription factor 2 (Runx2) and BMP2. When miR-204 expression is upregulated, osteoblast differentiation and function are suppressed, leading to reduced bone formation [69]. Regarding osteoclasts, miR-223 can regulate osteoclastogenesis and osteoclast activity. In PD patients, inflammatory infiltration alters miR-223 expression, thereby affecting osteoclast function and disrupting the balance between bone resorption and bone formation [70]. In the bone-brain axis of PD, some miRNAs can simultaneously impact both the nervous and skeletal systems [71], potentially indirectly affecting bone metabolism by influencing neuroendocrine pathways. In the brain, the hypothalamic-pituitary-adrenal (HPA) axis is closely related to bone metabolism. miRNAs can regulate the synthesis and secretion of HPA axis-related hormones, subsequently influencing the function of cells in bone tissue.

6.5. MiRNA therapy strategy

6.5.1 Targeted regulation of miRNAs

For miRNAs that are dysregulated and play critical roles in the PD bone-brain axis, drugs can be designed to regulate their expression. For instance, for miRNAs that are downregulated in PD and have neuroprotective effects on neurons, nucleic acid-based drugs can be developed to increase their levels in the body. Antisense oligonucleotides (ASOs) or microRNA mimics can be designed to enhance the expression of specific miRNAs [132]. Conversely, for miRNAs that are overexpressed in PD and promote disease progression, miRNA inhibitors such as antagomirs can be used to reduce their expression levels, thereby mitigating their adverse effects on neurons and bone cells [133].

6.5.2 Combination therapy strategy

Combine miRNA-targeted therapy with existing PD treatment methods. For example, while using dopamine replacement therapy to treat motor symptoms in PD patients, regulate miRNAs related to the bone-brain axis to improve patients' bone health and delay neurodegeneration. This combined treatment strategy holds promise for enhancing the overall quality of life for PD patients.

6.5.3 Development of biomarkers

Due to the specific expression changes of miRNAs in the PD bone-brain axis, they can be utilized as biomarkers for early diagnosis and prognosis assessment of PD. By detecting the levels of specific miRNAs in blood, cerebrospinal fluid, or tissue samples, it can assist in the diagnosis of PD, especially in the early stages of the disease where traditional diagnostic methods may not be sufficiently sensitive. Furthermore, monitoring the dynamic changes of relevant miRNAs during the treatment of PD patients can evaluate treatment efficacy and predict disease progression, providing a basis for clinicians to adjust treatment plans [134, 135].

Reviewer 2 Report

Comments and Suggestions for Authors

This is an interesting review article about the potential changes in bone metabolism in patients with Parkinson's disease (PD). For several years PD has been recognized as a systemic disease with its main clinical expression derived from progressive loss of dopaminergic neurons in the midbrain substantia nigra. This review article points out that clinicians managing PD patients need to be aware of PD effects on bone, and potential bone metabolism effects on CNS inflammation, which now appears to be a major mechanism of neurodegeneration in PD and other conditions.

I enjoyed reading this review. However, the review is compromised by both several poorly referenced statements and other paragraphs, which were well written and truthful, but had no obvious relevance to PD. I include the following:

“The decline in dopamine levels in Parkinson's patients may lead to disordered bone metabolism, further increasing the risk of osteoporosis.” (not referenced)

“Patients with PD may experience fluctuations in hormone levels, such as 160 changes in sex hormones and thyroid hormones, which directly impact bone formation 161 and resorption, thereby influencing bone metabolism.” (not referenced)

“Studies have indicated that chronic inflammation and immune system dysregulation can affect bone metabolism through various mechanisms, leading to osteoporosis or other bone problems. (not referenced)

 These abnormalities are diagrammed in Fig 2 which needs references.

 ER- not defined (suspect Estrogen or Estrogen Receptor)

4.2 5-HT 286 

“5-HT is a neurotransmitter widely distributed in the human body, playing a crucial 287 role in regulating mood, sleep, appetite, and pain perception. It not only modulates mood 288 and sleep in the brain but also exerts significant effects within the skeletal system. By binding to specific receptors on the surface of bone cells, 5-HT participates in the regulation of bone metabolism [63]. Studies have found that 5-HT can promote the proliferation and differentiation of osteoblasts while inhibiting the formation and activity of osteoclasts, thereby facilitating bone formation and inhibiting bone resorption to a certain extent. Furthermore, 5-HT is closely associated with the occurrence and progression of osteoporosis, and its mechanism of action in bone metabolism is increasingly becoming a research focus [64]. Recent research indicates a correlation between 5-HT and PD. Specifically, 5-HT interacts with the dopamine system in the brain, and dysfunction of the dopamine system is one of the primary pathological features of PD [65]. 5-HT indirectly regulates motor control and emotional responses by influencing the release and reuptake of dopamine.” This is a weak paragraph. No evidence presented how PERIPHERAL 5HT affects bones in PD.

4.3. NE 

“NE is an important neurotransmitter and hormone, and its role in the bone-brain axis cannot be overlooked. NE regulates bone metabolism primarily by binding to β-adrenergic receptors on the surface of bone cells [66]. Studies have shown that NE can promote osteoblast activity, increase bone matrix synthesis, while inhibiting osteoclast activity and reducing bone resorption. Additionally, NE regulates the differentiation direction of bone marrow mesenchymal stem cells by influencing the bone marrow microenvironment, thereby playing a significant role in bone formation and repair processes [67]. Furthermore, NE indirectly affects bone metabolism by influencing inflammatory and oxidative 309 stress responses [68]. NE is a crucial neurotransmitter that plays a key role in stress responses. When the human body is under pressure, the secretion of NE increases, triggering physiological responses such as accelerated heart rate and elevated blood pressure. Research indicates that the NE system may play a role in non-motor symptoms of PD, such as mood disorders, cognitive impairments, and sleep disturbances [69]. Moreover, noradrenergic neurons in certain regions of the brain may also be affected by the pathological processes of PD, further exacerbating patients' symptoms.” Same argument for NE as for 5HT above. No discussion of peripheral NE system in PD, which would be necessary for effects of NE on bone metabolism.

PTH- defined?

5.1. The Role of Inflammation in PD  This is a good section but has no relationship to bone metabolism in PD.

5.2. The Role of Immunity in PD Another very good section that has no obvious relationship to bone metabolism in PD

5.3. The Role of Inflammation in Bone Metabolism 5.4. The Role of Immunity in Bone Metabolism These are very good sections and are relevant to PD. Suggest eliminate 5.1 and 5.2

Figures in the text are helpful but need references in the Figure Legends.

There are a very few minor English errors that are easily corrected in final editing.

Author Response

This is an interesting review article about the potential changes in bone metabolism in patients with Parkinson's disease (PD). For several years PD has been recognized as a systemic disease with its main clinical expression derived from progressive loss of dopaminergic neurons in the midbrain substantia nigra. This review article points out that clinicians managing PD patients need to be aware of PD effects on bone, and potential bone metabolism effects on CNS inflammation, which now appears to be a major mechanism of neurodegeneration in PD and other conditions.

I enjoyed reading this review. However, the review is compromised by both several poorly referenced statements and other paragraphs, which were well written and truthful, but had no obvious relevance to PD. I include the following:

“The decline in dopamine levels in Parkinson's patients may lead to disordered bone metabolism, further increasing the risk of osteoporosis.” (not referenced)

Thank you for your valuable feedback and suggestion. Relevant references have been added.

“Patients with PD may experience fluctuations in hormone levels, such as 160 changes in sex hormones and thyroid hormones, which directly impact bone formation 161 and resorption, thereby influencing bone metabolism.” (not referenced)

Thank you for your valuable feedback and suggestion. Relevant references have been added.

“Studies have indicated that chronic inflammation and immune system dysregulation can affect bone metabolism through various mechanisms, leading to osteoporosis or other bone problems. (not referenced)

Thank you for your valuable feedback and suggestion. Relevant references have been added.

 These abnormalities are diagrammed in Fig 2 which needs references.

Thank you for your valuable feedback and suggestion. Relevant references have been added.

 ER- not defined (suspect Estrogen or Estrogen Receptor)

Thank you for your valuable feedback and suggestion. Defined on line 111.

4.2 5-HT 286

“5-HT is a neurotransmitter widely distributed in the human body, playing a crucial 287 role in regulating mood, sleep, appetite, and pain perception. It not only modulates mood 288 and sleep in the brain but also exerts significant effects within the skeletal system. By binding to specific receptors on the surface of bone cells, 5-HT participates in the regulation of bone metabolism [63]. Studies have found that 5-HT can promote the proliferation and differentiation of osteoblasts while inhibiting the formation and activity of osteoclasts, thereby facilitating bone formation and inhibiting bone resorption to a certain extent. Furthermore, 5-HT is closely associated with the occurrence and progression of osteoporosis, and its mechanism of action in bone metabolism is increasingly becoming a research focus [64]. Recent research indicates a correlation between 5-HT and PD. Specifically, 5-HT interacts with the dopamine system in the brain, and dysfunction of the dopamine system is one of the primary pathological features of PD [65]. 5-HT indirectly regulates motor control and emotional responses by influencing the release and reuptake of dopamine.” This is a weak paragraph. No evidence presented how PERIPHERAL 5HT affects bones in PD.

Thank you for your valuable feedback and suggestion. The relevant description of the effect of 5-HT on bone metabolism in PD has been added: Central 5-HT can regulate the secretion of pituitary hormones through neuroendocrine pathways, and these pituitary hormones in turn influence bone metabolism [78, 79]. When 5-HT levels decrease, this neuroendocrine regulatory mechanism is disturbed, potentially indirectly affecting bone metabolism. In the intestine, PD patients exhibit dysfunction of the enteric nervous system, and the intestine is the primary site of peripheral 5-HT synthesis [80]. Abnormalities in the enteric nervous system may affect the synthesis and release of 5-HT, subsequently influencing bone metabolism through the gut-bone axis [81]. Additionally, peripheral 5-HT can act on cells in bone tissue through blood circulation. When peripheral 5-HT levels or its signaling pathways change, they can indirectly affect bone metabolism [82].

4.3. NE

“NE is an important neurotransmitter and hormone, and its role in the bone-brain axis cannot be overlooked. NE regulates bone metabolism primarily by binding to β-adrenergic receptors on the surface of bone cells [66]. Studies have shown that NE can promote osteoblast activity, increase bone matrix synthesis, while inhibiting osteoclast activity and reducing bone resorption. Additionally, NE regulates the differentiation direction of bone marrow mesenchymal stem cells by influencing the bone marrow microenvironment, thereby playing a significant role in bone formation and repair processes [67]. Furthermore, NE indirectly affects bone metabolism by influencing inflammatory and oxidative 309 stress responses [68]. NE is a crucial neurotransmitter that plays a key role in stress responses. When the human body is under pressure, the secretion of NE increases, triggering physiological responses such as accelerated heart rate and elevated blood pressure. Research indicates that the NE system may play a role in non-motor symptoms of PD, such as mood disorders, cognitive impairments, and sleep disturbances [69]. Moreover, noradrenergic neurons in certain regions of the brain may also be affected by the pathological processes of PD, further exacerbating patients' symptoms.” Same argument for NE as for 5HT above. No discussion of peripheral NE system in PD, which would be necessary for effects of NE on bone metabolism.

Thank you for your valuable feedback and suggestion. The impact of peripheral NE system on PD bone metabolism has been increased. Due to dysfunction of the peripheral NE system in PD patients, the regulation of bone metabolism becomes imbalanced, leading to disruptions in the processes of bone formation and bone resorption [87]. In most cases, there is a relative increase in bone resorption and a relative decrease in bone formation, ultimately resulting in decreased bone mineral density. This reduction in bone mineral density significantly increases the risk of fractures in PD patients, severely impacting their quality of life and prognosis [88]. Besides decreased bone mineral density, the healing process after fractures in PD patients may also be affected by abnormalities in the peripheral NE system. Disturbances in the NE system may impact multiple aspects of fracture healing, including local inflammatory responses, cell proliferation and differentiation, and angiogenesis at the fracture site. For example, NE may interfere with the normal functional coordination between osteoblasts and osteoclasts at the fracture site, impeding the formation and remodeling of bone callus, thereby causing delayed fracture healing [89-91].

PTH- defined?

Thank you for your valuable feedback and suggestion. Defined on line 110.

5.1. The Role of Inflammation in PD  This is a good section but has no relationship to bone metabolism in PD.

Thank you for your valuable feedback and suggestion. 5.1 has been deleted.

5.2. The Role of Immunity in PD Another very good section that has no obvious relationship to bone metabolism in PD

Thank you for your valuable feedback and suggestion. 5.2 has been deleted.

5.3. The Role of Inflammation in Bone Metabolism 5.4. The Role of Immunity in Bone Metabolism These are very good sections and are relevant to PD. Suggest eliminate 5.1 and 5.2

Thank you for your valuable feedback and suggestion. 5.1 and 5.2 has been deleted.

Figures in the text are helpful but need references in the Figure Legends.

Thank you for your valuable feedback and suggestion. Relevant references have been added.

There are a very few minor English errors that are easily corrected in final editing.

Thank you for your valuable feedback and suggestion. Check the grammar of the manuscript.

Round 2

Reviewer 1 Report

Comments and Suggestions for Authors

The authors have revised their manuscript accordingly. No more comments and suggestions. 

Reviewer 2 Report

Comments and Suggestions for Authors

The authors have extensively re-written their paper and have addressed my prior criticisms. Specifically, they have added text related to the effects of 5HT and NE on bone metabolism in PD, have added a nice discussion of micro-RNA's in several places (I note revision of Fig3 to include miRNA's), have added missing references and eliminated the prior sections 5.1 and 5.2. There do not appear to be any English grammar concerns, which were minor in the original version. I feel that this paper is now ready for publication.